# Neuroprotective Effect of Eco-Sustainably Extracted Grape Polyphenols in Neonatal Hypoxia-Ischemia

**DOI:** 10.3390/nu14040773

**Published:** 2022-02-12

**Authors:** Hélène Roumes, Stéphane Sanchez, Imad Benkhaled, Valentin Fernandez, Pierre Goudeneche, Flavie Perrin, Luc Pellerin, Jérôme Guillard, Anne-Karine Bouzier-Sore

**Affiliations:** 1CRMSB, UMR 5536, University of Bordeaux and CNRS, F-33000 Bordeaux, France; helene.roumes@rmsb.u-bordeaux.fr (H.R.); stephane.sanchez@rmsb.u-bordeaux.fr (S.S.); imad.benkhaled@rmsb.u-bordeaux.fr (I.B.); valentin.fernandez@live.fr (V.F.); pierregoudeneche@gmail.com (P.G.); 2I3M, Common Laboratory CNRS-Siemens, University of Poitiers and Poitiers University Hospital, F-86073 Poitiers, France; 3IC2MP, UMR 7285, Team 5 Chemistry, University of Poitiers and CNRS, F-86000 Poitiers, France; flavie.perrin@univ-poitiers.fr; 4IRMETIST, Inserm U1313, University of Poitiers and CHU Poitiers, F-86021 Poitiers, France; luc.pellerin@univ-poitiers.fr

**Keywords:** neonatal hypoxia-ischemia, maternal supplementation, neuroprotection, resveratrol, grape-polyphenol, MRI, behavior

## Abstract

Polyphenols are natural compounds with promising prophylactic and therapeutic applications. However, their methods of extraction, using organic solvents, may prove to be unsuitable for daily consumption or for certain medical indications. Here, we describe the neuroprotective effects of grape polyphenols extracted in an eco-sustainable manner in a rat model of neonatal hypoxia-ischemia (NHI). Polyphenols (resveratrol, pterostilben and viniferin) were obtained using a subcritical water extraction technology to avoid organic solvents and heavy metals associated with chemical synthesis processes. A resveratrol or a polyphenol cocktail were administered to pregnant females at a nutritional dose and different time windows, prior to induction of NHI in pups. Reduced brain edema and lesion volumes were observed in rat pups whose mothers were supplemented with polyphenols. Moreover, the preservation of motor and cognitive functions (including learning and memory) was evidenced in the same animals. Our results pave the way to the use of polyphenols to prevent brain lesions and their associated deficits that follow NHI, which is a major cause of neonatal death and disabilities.

## 1. Introduction

Neonatal hypoxia-ischemia (NHI) is the second major cause of neonatal death worldwide, just behind preterm birth complications [1], and a major cause of chronic disability in full-term newborn infants (1–6 for 1000 births) [2,3]. Despite advances in neonatal intensive care, approximately 20% of these children die during the newborn period and an additional 25% exhibit irreversible motor and cognitive deficits [4,5]. The only therapeutic treatment consists of moderate hypothermia to be initiated within 6 h post-hypoxia-ischemia either by head or total body cooling. However, this “standard” care procedure (1) is inefficient in nearly 50% of patients [6] and (2) does not lead to a statistically significant reduction in neonatal mortality in low- and middle-income countries [7]. NHI is characterized by a disturbance in cerebral blood flow, which induces a deprivation of oxygen and energy substrates in the brain of the newborn, responsible for cell death and, more specifically, neuronal loss. For a long time, glucose has been considered as the only adequate brain energy substrate. Since the introduction of the astrocyte-neuron lactate shuttle (ANLS) concept by Pellerin and Magistretti in 1994 [8], lactate is considered as a prominent neuronal energetic fuel. Indeed, in this shuttle, glucose will be first metabolized by astrocytes (whose location is ideal between blood vessels and neurons) in lactate which will be transferred to neurons to be used as an energy source. While a lactate shuttle may support physiological brain activity, one could wonder if during/after a hypoxic-ischemic event, lactate *per se* may compensate for the glucose deficit. Indeed, in a previous study, we have shown that lactate administration post-NHI was neuroprotective [9]. In parallel to exploring this potential therapeutic possibility, we aimed at developing a preventive, nutritive and intergenerational neuroprotective approach. Therefore, the objective of this work was to determine whether maternal supplementation with polyphenols, administered during the last period of pregnancy, could reduce brain damages caused by NHI. Indeed, several studies, carried out on animals, have shown that the nutritional habits of the pregnant mother could have repercussions, in particular on the severity of brain lesions induced by NHI [10,11,12,13]. More recently, some studies have highlighted a neuroprotective role for resveratrol (RSV, direct i.p. injections to pups), a polyphenol present in grapes, in the context of NHI, via a reduction in brain lesion volume and the prevention of cognitive deficits [11,14,15]. Furthermore, we have shown that RSV (Figure 1) maternal supplementation (intergenerational approach) at nutritional doses (0.15 mg/kg/day in the drinking water of the pregnant female vs. 20–100 mg/kg i.p. injections in pups in the previous cited studies) was neuroprotective in the context of NHI [16]. In addition to an activation of the “classical” antioxidant, anti-apoptotic and anti-inflammatory pathways, we found that neuroprotection was also due to the modulation of brain energy metabolism via an upregulation of key partners of the ANLS. Similarly, another polyphenol, piceatannol (PIC, a more hydroxylated analogue of RSV, Figure 1), was also shown to be neuroprotective in NHI through the same intergenerational and nutritional approach [17]. Data indicated that PIC, whose bioavailability is higher compared to RSV, was more neuroprotective than RSV [18]. Therefore, polyphenolic bioavailability seems to be a key point of ensuring the best neuroprotection and increasing the success of translation to humans. Pterostilbene (PTE, Figure 1), an analogue of RSV, also present in grapes but to a lesser extent, has 2 methoxy groups compared to RSV, which confers to PTE a higher lipophilic character and therefore increases its bioavailability [19]. In the context of NHI, a single dose of 50 mg/kg PTE (i.p. injection in pups, 30 min before the hypoxic-ischemic event) was sufficient for obtaining antioxidant and anti-inflammatory effects [20]. Zeng et al. showed that administration of PTE (50 mg/kg), via oral gavage from postnatal day 3 (P3) to P8 pups was also neuroprotective [21]. A dimer of RSV, trans ε-viniferin (VNF, Figure 1) also found in grapes, exhibited neuroprotective properties in the context of neurodegenerative diseases (weekly i.p. injection, 10 mg/kg, in transgenic APPswePS1dE9 mice, from 3 until 6 months of age [22]). To our knowledge, the effects of these two polyphenols in on NHI have never been studied.

The objective of this work was to evaluate and compare the neuroprotective effect of RSV, PTE and VNF in NHI, through a nutritional (very low doses) and maternal (inter-generational approach) supplementation, using an eco-sustainable technique to extract these polyphenols. Moreover, a cocktail of polyphenols was tested, to demonstrate a potential synergistic effect when RSV is used in a mixture together with VNF and PTE, as demonstrated in vitro on a Parkinson’s disease model [23].

## 2. Materials and Methods

Chemistry. Grape canes of the variety “Ugni” (Vitaceae) were collected from Cognac’s vineyards in France. They were placed in an extruder (Clextral Evolum EV 53, Clextral, Firminy, France) apparatus. For extraction, as solvent we used a mixture of water and ethanol (2/8). After filtration, the solution was concentrated (40 °C, under vacuum condition) and finally lyophilized. The obtained lyophilized extract (4 g) was resuspended in a Biosolvants 1 mixture (20 mL; Makigreen D10-EtOAc-EtOH-H2O; 4:2:3:2). Then, it was injected into a Centrifugal Partition Chromatography (CPC) device. In our experimental conditions, after mixing the solvents at room temperature, two phases were obtained. After separation, the stationary phase (being the lower aqueous phase) was used to fill first the column prior to centrifugation at 1000 rpm. The mobile phase was further pumped into the column in ascending mode at a flow rate necessary for the separation (3 or 4 mL/min). When the mobile phase came out of the column, indicating that the hydrodynamic equilibrium was reached, the sample (dissolved in the stationary phase) was injected and fractions were collected and combined based on the High Performance Liquid Chromatography (HPLC) spectra, giving rise to three fractions of interest. All of the compounds were >95% pure by HPLC analysis. RSV came out in fraction 1 (101 mg), whereas VNF was present in fraction 2 (175 mg), together with other compounds. Fraction 2 was purified again by semi-preparative HPLC and the isolated yellow solid was characterized by ^1^H and ^13^C-Nuclear Magnetic Resonance (NMR) spectroscopy analysis in acetone-d6 and mass spectrometry. It was identified as VNF pure at 98%. VNF: Mp = 230 °C, NMR ^1^H (400 MHz, Acetone-d6) δ 8.36 (s, 2H), 8.18 (s, 1H), 7.19 (dd, J = 12.8, 8.6 Hz, 4H), 6.91 (d, J = 16.3 Hz, 1H), 6.86–6.81 (m, 2H), 6.72 (dd, J = 12.4, 9.1 Hz, 4H), 6.33 (d, J = 2.0 Hz, 1H), 6.24 (s, 3H), 5.42 (d, J = 5.4 Hz, 1H), 4.48 (d, J = 5.4 Hz, 1H). NMR ^13^C (400 MHz, dimethyl sulfoxide (DMSO)-d6) δ 160.7, 158.7, 158.6, 157.8, 157.5, 145.5, 134.8, 131.5, 128.9, 127.7, 127.6, 127.0, 121.8, 118.2, 115.6, 115.2, 105.4, 103.1, 101.1, 95.9, 92.4, 55.1. Mass Spectrometry-Fast Atom Bombardment (MS-FAB) m/z 455 [M + 1] 455.

Animals and Nutritional Maternal Supplementation. All of the animal procedures were conducted in accordance with the Animal Experimentation Guidelines of the European Communities Council Directive of November 24, 1986 (86/609/EEC). Protocols met the ethical guidelines of the French Ministry of Agriculture and Forests and were approved by the local ethics committee (CEEA50-ethical approval code 9476). Pregnant Wistar female rats (Janvier Laboratories, France) were received 1 week before delivery (15 days of gestation), kept on light/dark 12 h/12 h cycle. Water was available *ad libitum* and pregnant females were fed with A03 food (SAFE, Augy, France). For the supplemented condition, a cocktail of polyphenols composed of RSV, PTE (gift from Exinnov, Saint Jean d’Illac, France) and VNF was added to the drinking water of dams for 2 weeks (last week of gestation + first week of breastfeeding; equimolar doses: 0.15 mg/kg/d for RSV and PTE and 0.30 mg/kg/d for VNF). A measurement 16 µL of absolute ethanol were added into the drinking water (200 mL) for the hypoxic-ischemic and sham groups for 2 weeks (last week of gestation + first week of breastfeeding). Five experimental groups were established according to the maternal supplementation plan and occurrence or not of a hypoxic-ischemic event: sham group (pups without maternal supplementation and without NHI); HI group (pups without maternal supplementation and with NHI); RSV-HI group (maternal supplementation with RSV, during two weeks before NHI); Cocktail-HI group (maternal supplementation with the polyphenolic cocktail, during two weeks before NHI) and HI-Cocktail group (NHI and maternal supplementation with the polyphenolic cocktail, during two weeks after NHI).

Hypoxia-Ischemia Model. NHI was induced as previously described [9]. Briefly, P7 rat pups of both genders underwent ischemia via a permanent ligation of the left common carotid artery. After surgery and a 30-min recovery period, pups were placed in a hypoxic chamber (Intensive Care Unit Warmer, Harvard Apparatus, Les Ulis, France, 2h, 8% O_2_, 92% N_2_).

Longitudinal Magnetic Resonance Imaging Acquisition. Diffusion Magnetic Resonance Imaging (MRI) acquisitions were performed 3 h after the carotid artery ligation (P7) for the sham, HI, RSV-HI and Cocktail-HI groups, then 48 h after (P9) and 23 days later (P30) for all groups plus the HI-Cocktail group (as illustrated in Figure 2 below), on a horizontal 4.7 T Biospec 47/50 system (Bruker, Ettlingen, Germany) equipped with a 6 cm BG6 gradient system (1000 mT/m).

Pups were anesthetized with isoflurane (4% for induction and 1.5% for maintenance) and body temperature was maintained at 35.5 ± 0.5 °C during acquisition with a water-heated MRI bed. Brain lesion volumes and edema severity were assessed by Diffusion Weighted Imaging (DWI): 20 axial slices (0.7 mm thick), 30 directions, b-value 1000 s/mm^2^, echo time (TE) = 24 ms, repetition time (TR) = 2 s, Δ = 8.11 ms, δ = 2.5 ms, total duration = 17 min 04 s. MRI Analysis. Measurements were performed with Paravision 6.0.1 software (Bruker BioSpin, Karlsruhe, Germany). For the different time points and for each pup, the brain damage area as well as the total brain surface were delineated on the 20 adjacent slices of the DWI (slice thickness: 0.7 mm). Lesion volumes were expressed as a percentage of the total brain volume. At P7, apparent diffusion coefficient (ADC) values (mm^2^/s) were measured in the cortex, hippocampus, and striatum, on the ADC maps resulting from DWI acquisition, as previously described [16]. For MRI analyses, lesion volumes and ADC were measured as previously described [18].

Behavioral Tests. Behavioral tests were performed as previously described and blindfolded [17]. Righting Reflex (P8, P10, P12). Pups were placed on the back and time to turn over on their 4 paws was measured. Data are the mean of 3 trials. mNSS (P24). Severity of the neurological damage was evaluated by a series of motor, sensory and reflex tasks. If the animal fails to perform the task, 1 point is awarded, while no point is given if successful (Total score: 0 to 1: no impairment; 2 to 6: moderate impairments; 7 to 12: impairments; 13 to 18: severe impairments). Novel Object Recognition Test (P42–P45). Long-term, hippocampal-dependent memory was assessed as previously described [9]. Briefly, after a 1st exploration of the empty open space on the 1st day (exploration: 5 min), on the 2nd day, an exploration in the presence of 2 identical objects was performed (exploration: 5 min). The next day, after one of the 2 old objects had been replaced by a new object, a new exploration was performed (exploration: 5 min). Durations of each object exploration were recorded. The discrimination I ratio was defined as: I = (time spent on new object-time spent on familiar object)/(time spent on new object + time spent on familiar object).

Statistical Analysis. Statistical analyses were performed using GraphPad Prism 7.00 software. The same software was used to create graphs. All data were expressed as mean ± standard error of the mean (s.e.m). Number of pups is indicated within graphs. Statistical significance of the differences between multiple groups was determined using One-Way ANOVA and Fischer’s LSD post hoc test.

All study data are included in the main text.

## 3. Results

### 3.1. Chemistry

Grape polyphenols are usually extracted from plant material (grape canes and roots) using traditional organic solvent methods due to their strong polarities. Solvents such as ethanol, methanol, acetone, and ethyl acetate have been widely used in conventional solid-liquid extraction processes. However, the excessive use of organic solvents generates toxic wastes and has therefore harmful environmental and human health impacts. In addition, traditional extraction methods are both time consuming and labor intensive. For this reason, an eco-efficient process was designed to avoid the use of organic solvents and repeated extraction procedures. This eco-efficient process is based on the use of reactive subcritical water extrusion in a Clextral Evolum EV 53 extruder (Figure 1).

The subcritical water extraction technology is very attractive and efficient. Its advantages are shorter extraction times, high aselectivity, and the ease of removing the solvent after extraction by simple decompression. Water becomes subcritical at temperatures above 100 °C but below the critical point (374 °C) and at pressures below 221 bars. Water then remains in a liquid state but is transformed into a solvent with particularly interesting properties in the field of plant extraction. Under these conditions, the viscosity and surface tension of subcritical water are lower than those of water at room temperature, its diffusivity increases and its dielectric constant becomes close to that of organic solvents such as ethanol or acetone. This results in a better penetration of the solvent into the plant matrix, an improved mass transfer and a faster diffusion rate, thus a higher extraction yields with very short contact times.

Grape polyphenols used in this study were obtained by reactive extrusion from grape canes of Ugni blanc varieties, specific to the Cognac country. The concentrated aqueous extract obtained exhibits the HPLC profile shown in Figure 2, with mass percentages for RSV of 4.9% and for VNF of 10.8%, but also other known oligomers of the stilbenoid family (ampelopsin, vitisin A and B).

For the purification of our compounds of interest, CPC has been implemented on a pilot scale. While preparative HPLC and flash chromatography rely on a solid silica stationary phase, CPC does not contain silica and uses two immiscible liquids as stationary and mobile phases. The separation of our target molecules was based on their respective affinities with the liquid phases, expressed by the partition coefficient (KD). The CPC technique allowed to obtain RSV and VNF in a reliable and reproducible manner. An additional innovation in this extraction process was the use of sustainable solvents such as ethanol, ethyl acetate and water.

### 3.2. Comparison of the Neuroprotective Effects between RSV and Cocktail Maternal Supplementation for Two Weeks before the HI Event in Neonates

NHI induced rapid brain damages in brain pups (hypointense signal due to tissue edema, Figure 3A). Brain volume lesions were measured at P7, 3 h after the common carotid artery ligation (Figure 3B). Pups in the Cocktail-HI group presented smaller lesion volumes than pups in the HI and RSV-HI groups (44 ± 2%, 20 ± 5% and 33 ± 2% for HI, Cocktail-HI and RSV-HI groups, respectively). HI induces cerebral edema, whereby the severity was evaluated by the ADC; the smaller the ADC, the more severe the edema (Figure 3C). In the striatum, ADC values were the lowest for pups in the HI group, indicating a higher severity of edema. Interestingly, compared to RSV, maternal supplementation with the cocktail allowed a higher striatal neuroprotection (ADC value in the Cocktail-HI group higher compared to values in the HI group). In addition to MRI, behavioral tests were performed (Figure 3D) to evaluate the impact of the neuroprotection observed on MR images on neurological functions. Neurological impairments were assessed using the modified Neurological Severity Score (mNSS). The mNSS score was highest for pups in the HI group, compared to pups in the supplemented groups, indicating more severe neurological damages for pups whose mother was not supplemented. Moreover, no difference in the mNSS scores was found between sham and Cocktail-HI groups, which correlates with a highest striatal neuroprotection observed by MRI in the Cocktail-HI group (Figure 3A, C + S).

### 3.3. Can Maternal Polyphenolic Supplementation Be Also Curative?

To evaluate the potential curative effects of maternal supplementation, the polyphenolic cocktail was administered in the drinking water of the breastfeeding females just after the HI event in pups (HI-Cocktail group). Brain lesion volumes were evaluated by diffusion MRI at P9 for all groups (Figure 4A), which corresponded to 48 h of maternal supplementation for the HI-Cocktail group. Brain lesion volumes of pups in the HI-Cocktail group were smaller than those of the pups in the HI group (19 ± 4% and 27 ± 3% for HI-Cocktail and HI groups, respectively). At this early time point, pups in the Cocktail-HI and RSV-HI groups presented the smallest lesion volumes. At the corresponding time point, behavioral experiments were performed. Early reflexes of pups were evaluated from P8 to P12 (Figure 4B). During righting reflex, whatever the time point, pups of the HI group presented a delay in the accomplishment of the task (Figure 4C). Such a delay was detected for the HI-Cocktail group until P10. From P8 to P12, Cocktail-HI group performed as well as the sham group. At P12, abilities of pups in cocktail-supplemented groups were better than that of pups in the RSV group.

### 3.4. Is the Neuroprotection Still Effective over Time?

The lasting nature of neuroprotection was evaluated by diffusion MRI at P30 (Figure 5A). At P30, brain lesion appears as a hypersignal (tissue loss). Brain lesion volumes were significantly reduced in the supplemented groups, compared to the HI group (28 ± 5%, 4 ± 3%, 8 ± 5% and 10 ± 3% for HI, Cocktail-HI, HI-Cocktail and RSV-HI groups, respectively) (Figure 5B). Remarkably, hippocampi of supplemented groups were more preserved than that of the HI group. Hippocampal-dependent long-term memory was therefore tested for all groups using the novel object recognition test. For the HI group, HI induced an impairment of long-term memory which was completely counteracted by maternal supplementation for all the supplemented groups, compared to the sham group (Figure 5C).

## 4. Discussion

Over the last few years, a growing number of studies attempted to demonstrate a protective effect of polyphenols against brain damages. RSV is the polyphenol that has received the most attention [11,14,15]. Although antioxidant and anti-inflammatory properties of this polyphenol have been highlighted, several limitations can be underlined, such as the need of high doses to obtain beneficial effects (ranging from 20 to 100 mg per day) and the use of intraperitoneal injection, which is known to be pro-inflammatory. Moreover, the commercial source of RSV, obtained by chemical synthesis, can be possibly contaminated with catalyst residues (heavy metals) and traces of organic solvents. To overcome these limitations, in this study polyphenol supplementation to animals was (i) given to nutritional doses (0.15 mg/kg/d), (ii) administered via maternal supplementation in the drinking water, and (iii) using polyphenols from a green and eco-sustainable extraction technique.

Under similar conditions, nutritional maternal supplementation with RSV or PIC was previously demonstrated to be neuroprotective by (i) decreasing brain lesion volumes, (ii) decreasing brain edema, (iii) limiting neuronal death, and (iv) preserving motor and cognitive functions [16,17,18]. Even under our low doses of polyphenols, and using our intergenerational approach, the neuroprotection was efficient for the pups which underwent a hypoxic-ischemic episode. Moreover, in this study we have shown that the polyphenolic cocktail allowed for a better neuroprotection of pups’ striatum than maternal supplementation with RSV alone, indicating a synergistic effect when RSV is used in a mixture together with VNF and PTE, which could be explained by an increase in bioavailability. This higher neuroprotection was evidenced by the striatal ADC values (less edema severity), obtained from DWI and interestingly it was correlated with better abilities of pups in the mNSS test, which is particularly relevant to highlight the differences in striatal neuroprotection. Indeed, it has been shown that acute impairments in the functional performances evaluated by the mNSS correlate with striatal infarct volume [24]. Since the striatum is a brain region highly vulnerable in human newborns and is very sensitive to hypoxia [25,26], the enhanced neuroprotective effect of the cocktail on this brain structure is of particular interest. It has been shown that a perinatal striatal injury can have deleterious consequences in adulthood by being responsible for the decreased number of neurons of the substantia nigra, which is known to receive trophic support from the striatum [27]. In addition, brain lesions that can damage the development of cortico-striatal circuits lead to neuropsychological dysfunction, such as an increased risk of attention deficit hyperactivity disorder and/or other disruptive behavioral problems [28].

Interestingly, in addition to a neuroprotective effect of maternal supplementation before the hypoxic-ischemic event with the polyphenolic cocktail, the cocktail also showed a beneficial effect when it was administered in the drinking water of the breastfeeding females, i.e., after the hypoxic-ischemic event in the pups, indicating curative properties. Results from the HI-Cocktail group demonstrated that only 2 days of maternal polyphenolic supplementation was sufficient to get a beneficial effect. Indeed, brain lesion volumes at P9 in the HI-Cocktail group were already smaller compared to the HI group. Concerning behavior, surprisingly, the righting reflex, which allow to evaluate early reflexes (between P8 and P12) indicated that pups in the HI-Cocktail group were able to recover to the same level compared to pups whose mother was supplemented two weeks before the hypoxic-ischemic event. After only 5 days of maternal supplementation with the cocktail, pups in the HI-Cocktail group performed as well as the one in the Cocktail-HI group (no statistical difference). Moreover, righting reflexes of pups in the RSV-HI group at P12 were not as good as those of the pups whose mother had been supplemented with the cocktail (both Cocktail-HI and HI-Cocktail), confirming the hypothesis of synergistic properties given by the cocktail. Maternal polyphenol supplementation was therefore neuroprotective following preventive administration (before hypoxic-ischemic event in the pups, Cocktail-HI group) but also beneficial during curative administration (after hypoxic-ischemic event in the pups, HI-Cocktail group).

Then, the question of the lasting neuroprotection of these supplementations arose. At P30, the diffusion MR images showed a hippocampal neuroprotection for all of the supplemented groups. No difference in hippocampal neuroprotection could be measured between the different supplementation groups in the diffusion MR images. To evaluate the functional consequences of this hippocampal neuroprotection, the novel object recognition test was performed, which probes hippocampal-dependent long-term memory [9]. MRI data were corroborated by our cognitive data since NHI induced an impairment of hippocampal-dependent long-term memory, which was prevented by polyphenolic maternal supplementation and no difference was highlighted between all of the supplemented groups. Once again, very interestingly, when the maternal supplementation was administrated after the NHI (HI-Cocktail group), pups recovered the same levels of cognitive skills as the pups whose mothers were supplemented two weeks before the hypoxic-ischemic event, or, more importantly, as the pups in the sham group, indicating a curative effect that lasted as well for a prolonged period of time.

This study is the first one that used green chemistry extracted polyphenols, administered at nutritional doses, via pregnant and/or breastfeeding females, to determine the ability of these polyphenols to prevent or counteract brain lesions due to NHI. In addition, the comparison with RSV alone or in a mixture with two other polyphenols (PTE + VNF) is innovative since such a cocktail has never been studied before for this biological application. These innovative results could be translated into human application by advising to take food supplements during pregnancy (preventive approach) or during breastfeeding (curative approach); these food supplements were composed of a polyphenolic cocktail. The possibility of a prophylactic use in humans would be facilitated by the fact that a food supplement, based on eco-sourced natural products, does not have to meet the same safety requirements that drugs must meet to be put on the market. Altogether, our data highlight a promising perspective for nutraceutical therapeutic development in the context of NHI, the second major cause of neonatal death worldwide, for which no pharmacological treatment exists.

## Data Availability

All data are presented in the article, are original and not published elsewhere.

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
