# Peer review of "Neuroprotective Effect of Eco-Sustainably Extracted Grape Polyphenols in Neonatal Hypoxia-Ischemia"

_nutrients, 2022, doi:10.3390/nu14040773_

Round 1

Reviewer 1 Report

A well designed experimental study fulfilling all the requirements for animal studies.

Results are clearly presented and well documented.

All abbreviations need definition.

I would like to see in the conclusions section a comment on the authors’opinion as to how their results could be translated in humans

Author Response

Reviewer 1

A well designed experimental study fulfilling all the requirements for animal studies.

Results are clearly presented and well documented.

We would like to thank the reviewer for his comments.

All abbreviations need definition.

All abbreviations are now defined in the text.

I would like to see in the conclusions section a comment on the authors’opinion as to how their results could be translated in humans

The reviewer's comment is well taken. Thus, the following paragraph was added in the conclusion:

These innovative results could be translated into human application by advising to take food supplements during pregnancy (preventive approach) or during breastfeeding (curative approach); these food supplements being composed of a polyphenolic cocktail. The possibility of a prophylactic use in humans would be facilitated by the fact that a food supplement, based on eco-sourced natural products, does not have to meet the same safety requirements that drugs must meet to be put on the market.

Reviewer 2 Report

Journal: Nutrients

Manuscript ID: Nutrients-1571611

Neuroprotective effect of eco-sustainably extracted grape polyphenols in neonatal hypoxia-ischemia

Hélène Roumes , Stéphane Sanchez , Imad Benkhaled , Valentin Fernandez , Pierre Goudeneche , Flavie Perrin , Luc Pellerin , Jérôme Guillard * , Anne-Karine Bouzier-Sore

I thank the Editor and the Editorial Board members for providing me an opportunity to review this manuscript. I appreciate the authors for the research design and scientific soundness of the manuscript but then improve their presentation.

Reviewers’ comments

  1. Abstract, line 16 “However, their production and administration modes are inadequate for certain medical indications” Is it true that there are inadequate modes of administration? Be more specific or consider re-editing the sentence.
  2. The word long-term “beneficial study /effect” should be rewritten as it misleads the readers. The author studied for P30 and it is not appropriate to mention it as a long-term beneficial effect.
  3. The usage of abbreviations should be uniform and use the full-form on its first use.
  4. The mentioned NHI model whether induces hypoxia only in the brain or induces hypoxia in other tissues too. Whether the authors measured the polyphenol protective effect in the retina?
  5. Hypoxia-induced damage is uniform throughout different regions of the brain or it is specific to particular regions?
  6. Composition of polyphenols in ugni varieties before and after extraction by reactive subcritical water extrusion remains the same?
  7. Rewrite lines 145 to 149 as it is confusing. I recommend the experimental design with timelines be presented as a graphical illustration for better understanding. The timelines and group names are misleading it may be rewritten.
  8. Why E15 was selected for therapeutic feeding of polyphenol and why 30 mg/kg/d of VNF and only 0.15 mg/kg/d for RSV and PTE why equal dose concentrations were not used?
  9. In figure 2 the fonts and labeling should be clear.
  10. Figure 3A and 4A mention what is the black, white, and grey areas and label what we are looking for?
  11. Whether the lesion volume increases with time?
  12. Behavioral test methodology can be explained in detail. Whether tests are conducted blindfolded? MNS score was awarded manually or automatically calculated by software?
  13.  What are the female adult N per group and each group as to how many litters? what is the male and female pup ratio per group?
  14. Add how ADC values, brain lesion volume, and MNS score are calculated in methodology?
  15. I was surprised to see that the HI-Cocktail group in the experimental design showed that only 2 days of maternal polyphenolic supplementation with a lower nutritional dose (0.15 mg/kg/d) was sufficient to get a beneficial effect. After only 5 days of maternal supplementation with the polyphenol cocktail, pups in the HI-Cocktail group performed as well as the ones in the Cocktail-HI group (no statistical difference). Pre and post-treatment of the cocktail were almost the same why? Kindly comment on this

Author Response

Reviewer 2

I thank the Editor and the Editorial Board members for providing me an opportunity to review this manuscript. I appreciate the authors for the research design and scientific soundness of the manuscript but then improve their presentation.

We would like to thank the reviewer for considering our study and for helping to improve the manuscript.

  1. Abstract, line 16 “However, their production and administration modes are inadequate for certain medical indications” Is it true that there are inadequate modes of administration? Be more specific or consider re-editing the sentence.

We thank the reviewer for this remark. It is true that the sentence is imprecise and can lead to confusion. We have clarified our thoughts by changing to the following sentence: “However, their methods of extraction, using organic solvents, may prove to be unsuitable for consumption or for certain medical indications.”

  1. The word long-term “beneficial study /effect” should be rewritten as it misleads the readers. The author studied for P30 and it is not appropriate to mention it as a long-term beneficial effect.

We thank the reviewer for his fine proofreading. It is true that the term "long-term" can lead to confusion. We wanted to discriminate between very short-term effects, just after NHI (e.g. early reflexes P8-P12), and late effects, visible in young rats (P30-P45). To avoid misleading the reader, we have replaced the term "long-term" by the term "overtime". We hope to have addressed the comment appropriately and removed doubt on the period studied.

  1. The usage of abbreviations should be uniform and use the full-form on its first use.

All abbreviations are now defined and standardized in the text.

  1. The mentioned NHI model whether induces hypoxia only in the brain or induces hypoxia in other tissues too. Whether the authors measured the polyphenol protective effect in the retina?

We thank the reviewer for his very interesting question. Ligation of the common carotid artery induces a decrease in blood flow in the brain of the pups which is compensated by the polygon of Willis. To induce hypoxia-ischemia, this ligation must be completed by 2h in a hypoxia chamber at 8% O2. Hypoxia is therefore systemic. However, our study focused on the effects at the brain level and we did not explore the possible protective effects of polyphenols on the retina of the pups. The role of polyphenols in retinal protection is of growing interest and we hope to explore this aspect in future studies.

  1. Hypoxia-induced damage is uniform throughout different regions of the brain or it is specific to particular regions?

In our model, hypoxia-ischemia induces lesions of the ipsilateral hemisphere (approximately 40% of the total volume of the brain) but all the structures are not affected to the same extent by hypoxia-ischemia. When neuroprotection is measured, the cortex is weakly protected, while the hippocampus is less affected and the striatum is sometimes spared. These differences between structures can be highlighted by MRI thanks to the measurement of the apparent diffusion coefficient (ADC) values (mm2/s): the smaller the ADC, the more severe the edema.

  1. Composition of polyphenols in ugni varieties before and after extraction by reactive subcritical water extrusion remains the same?

The reactive extrusion technology is a non-destructive technique. It only allows to increase the extraction yield of low abundance molecules present in the plant, like grape canes. Thus, the composition of stilbenoids (polyphenols) before extrusion or after is the same. Only the effective yield of polyphenols has increased and we find at the end in the same proportions the same 5 desired molecules, i.e. Ampolopsin, Resveratrol, Viniferin, as well as Vitisins A and B.

  1. Rewrite lines 145 to 149 as it is confusing. I recommend the experimental design with timelines be presented as a graphical illustration for better understanding. The timelines and group names are misleading it may be rewritten.

In our version of the article, lines 145-149 correspond to:

“Longitudinal Magnetic Resonance Imaging Acquisition. Diffusion Magnetic Resonance Imaging (MRI) acquisitions were performed 3 h after the carotid artery ligation (P7) for the sham, HI, RSV-HI and Cocktail-HI groups, then 48 h after (P9) and 23 days later (P30) for all groups plus the HI-Cocktail group (as illustrated in Scheme 2 below), on a horizontal 4.7 T Biospec 47/50 system (Bruker, Ettlingen, Germany) equipped with a 6 cm BG6 gradient system (1000 mT/m).” As recommended by the reviewer, the experimental design with timelines was added and the group names were clarified.

  1. Why E15 was selected for therapeutic feeding of polyphenol and why 30 mg/kg/d of VNF and only 0.15 mg/kg/d for RSV and PTE why equal dose concentrations were not used?

In our study, E15 was selected for therapeutic feeding of polyphenols because this time period corresponds to the last trimester of gestation in humans. Polyphenols were administered at equimolar concentration. Since VNF as a molecular weight which is twice the one of RSV or PTE, than you need to administer a two-fold higher concentration on a weight basis.

  1. In figure 2 the fonts and labeling should be clear.

We thank the reviewer for his remark; the readability of Figure 2 has been improved.

  1. Figure 3A and 4A mention what is the black, white, and grey areas and label what we are looking for?

Clarifications have been made in the legends of Figures 3A and 4A to help understand the images:

Figure 3A: The lesion appears as a hyposignal, reflecting cerebral edema.

Figure 4A: At P9, the lesion consists of zones of still turgid cells which appear as a hyposignal and zones of apoptotic and necrotic cells which appear as a hypersignal.

  1. Whether the lesion volume increases with time?

In our model of HIN, brain lesion volumes decrease over time, even for the group not neuroprotected by maternal supplementation (HI group) due to recovery of the penumbra zone.

  1. Behavioral test methodology can be explained in detail. Whether tests are conducted blindfolded? MNS score was awarded manually or automatically calculated by software?

The reviewer is right, the behavioral tests were not properly explained. We have already detailed these tests in a previous study aimed at evaluating the effects of maternal piceatannol supplementation. To facilitate the reader's understanding, the reference has been added:

“Behavioral tests were performed as previously described and blindfolded [17]”

Yes, tests were performed blindfolded so it is also indicated. The quantification of mNSS was performed manually.

  1. What are the female adult N per group and each group as to how many litters? what is the male and female pup ratio per group?

Two female adults were considered per group and each litters contained 8 pups. Both male and female were considered for this study and no significant difference could be demonstrated between the two groups.

  1. Add how ADC values, brain lesion volume, and MNS score are calculated in methodology?

We have already detailed these tests in a previous study aimed at evaluating the neuroprotective effects of maternal polyphenol supplementation in a context of NHI combined with maternal alcoholism. To facilitate the reader's understanding, the reference has been added:

“For MRI analyses, lesion volumes and ADC were measured as previously described [18].”

  1. I was surprised to see that the HI-Cocktail group in the experimental design showed that only 2 days of maternal polyphenolic supplementation with a lower nutritional dose (0.15 mg/kg/d) was sufficient to get a beneficial effect. After only 5 days of maternal supplementation with the polyphenol cocktail, pups in the HI-Cocktail group performed as well as the ones in the Cocktail-HI group (no statistical difference). Pre and post-treatment of the cocktail were almost the same why? Kindly comment on this

The reviewer is right, the results we have obtained in pups via maternal polyphenol supplementation are very promising, both for the preventive approach (cocktail before the NHI) and for the curative approach (cocktail after the NHI).

As specified in the M&M, the composition of the cocktail is identical for all groups whose dams were supplemented with the cocktail (0.15 mg/kg/d for RSV and PTE + 0.30 mg/kg/d for VNF):

“Five experimental groups were established according to the maternal supplementation plan and occurrence or not of a hypoxic-ischemic event: sham group (pups without maternal supplementation and without NHI); HI group (pups without maternal supplementation and NHI); RSV-HI group (maternal supplementation with RSV, during two weeks before NHI); Cocktail-HI group (maternal supplementation with the polyphenolic cocktail, during two weeks before NHI) and HI-Cocktail group (NHI and maternal supplementation with the polyphenolic cocktail, during two weeks after NHI).”

The lack of difference between the HI-Cocktail and Cocktail-HI groups can be explained by the great cerebral plasticity of newborns. Indeed, at P9 (48 hours after HI and after only 48 hours of supplementation for the HI-Cocktail group), no significant difference could be demonstrated in the lesion volumes between these two groups. Similar results were measured at P30, for these two groups, both for lesion volumes and for the novel object recognition test. These results are in agreement with the results of a previous study testing the effects of piceatannol: no difference could be demonstrated between the preventive approach and the curative approach (maternal supplementation: 0.15 mg/kg/d in piceatannol) [Maternal consumption of piceatannol: A nutritional neuroprotective strategy against hypoxia-ischemia in rat neonates. Dumont U, Sanchez S, Olivier B, Chateil JF, Pellerin L, Beauvieux MC, Bouzier-Sore AK, Roumes H. Brain Res. 2019 Aug 15;1717:86-94.]
